# Diagnostic Value of the Five Times Sit-to-Stand Test and Other Functional Measures in Patients with Arteriosclerosis Obliterans

**DOI:** 10.3390/healthcare13222903

**Published:** 2025-11-14

**Authors:** Changsung Han, Gwon-Min Kim, Sung Woon Chung, ChungWon Lee, Miju Bae, Chiseung Lee, Up Huh

**Affiliations:** 1Department of Thoracic and Cardiovascular Surgery, Pusan National University College of Medicine, Biomedical Research Institute, Pusan National University Hospital, Busan 49241, Republic of Korea; 2Medical Research Institute, Pusan National University, Yangsan 50612, Republic of Korea; 3In Silico Medicine Lab, Biomedical Research Institute, Pusan National University Hospital, Busan 49241, Republic of Korea

**Keywords:** arteriosclerosis obliterans (ASO), functional assessment, five times sit-to-stand test (FTSS), six-minute walk test (SMWT), gait speed

## Abstract

**Background/Objectives:** Physical examination techniques are commonly used for the diagnosis and evaluation of Arteriosclerosis Obliterans (ASO). However, these methods do not objectively reflect the degree of claudication or functional impairment reported by patients. Moreover, standardized functional metrics to facilitate consistent clinical communication and prognostic assessments among healthcare providers are lacking. This study aimed to identify performance-based functional tests that enable quantitative assessment of symptoms and gait limitation in patients with ASO and to propose them as objective, reproducible clinical indicators. **Methods:** Fifty-six participants (27 patients with ASO and 29 healthy controls) underwent multiple functional tests, including the Five Times Sit-to-Stand Test (FTSS), six-minute walk test (SMWT), gait speed, Short Physical Performance Battery (SPPB), and grip strength. Test results were compared between the groups and evaluated against ankle-brachial index (ABI) values for diagnostic and functional relevance. **Results:** Patients with ASO demonstrated significantly lower SPPB scores, slower gait speed, longer FTSS times, and shorter SMWT distances than controls, whereas grip strength did not differ. Random forest and receiver operating characteristic analyses identified the FTSS, SMWT, and gait speed as significant predictors of ASO. **Conclusions:** The FTSS, SMWT, and gait speed are simple and clinically meaningful performance-based measures that can complement the ABI in the evaluation of ASO. Combining the FTSS with the SMWT and gait speed may provide a more comprehensive and reliable functional assessment and facilitate early screening, guide clinical decision-making, and enable objective evaluation of functional recovery before and after treatment, while improving patient self-assessment and communication among healthcare providers.

## 1. Introduction

Arteriosclerosis Obliterans (ASO) is a chronic, progressive vascular disease caused by atherosclerotic narrowing or occlusion of the arteries supplying the lower extremities. This commonly leads to functional limitations such as intermittent claudication. ASO is widely regarded as a representative vascular aging disease and its prevalence is strongly associated with advancing age. Epidemiological studies indicate that more than 20% of individuals aged 65 years or older are affected by ASO, and its prevalence is expected to increase markedly with the rapid demographic shift toward an aging society [1,2,3]. According to national demographic projections, the proportion of individuals aged 65 years and above in Korea is expected to triple by 2060 [4]. This trend underscores the urgent need for effective strategies for the early detection and functional assessment of vascular diseases in older adults.

Clinically, ASO is characterized by reversible muscular ischemia manifesting as cramp-like pain and aching sensations in the affected muscles during ambulation. These symptoms substantially impair the capacity for physical activity and walking performance, leading to a significant deterioration in overall physical function [5].

Although several diagnostic tools are currently used to assess the severity of ASO, such as peripheral pulse palpation, Doppler auscultation, ankle-brachial index (ABI) measurement, and computed tomography angiography, which are generally costly and time-consuming, these methods often fail to accurately capture subjective symptoms, particularly the degree of claudication or limitations in daily physical functioning [1,2,6]. In clinical practice, healthcare providers frequently rely on patients’ self-reported “pain-free walking distances” to estimate symptom severity. However, this measure is highly vulnerable to subjective bias and lacks standardized quantification, thereby limiting its value as an objective clinical indicator [7,8]. Such limitations often contribute to delayed diagnoses, causing patients to miss critical windows for timely therapeutic interventions. Furthermore, the absence of a standardized tool for evaluating the functional status hinders the consistent communication of clinical information across providers and impedes the accurate assessment of prognosis, treatment response, and functional decline in cases of disease recurrence [4,9,10].

To address these gaps, the present study implemented a comprehensive assessment of physical function in patients with ASO using multiple performance-based measures. Specifically, we evaluated gait speed, the six-minute walk test (SMWT), grip strength, the Short Physical Performance Battery (SPPB), and the Five Times Sit-to-Stand Test (FTSS), and conducted a comparative analysis across these functional metrics [11,12,13]. By employing these objective performance-based assessments, this study aimed to overcome the shortcomings of conventional diagnostic methods and provide a more reliable framework for evaluating functional impairment in patients with ASO.

## 2. Materials and Methods

### 2.1. Study Design and Participants

The participants were enrolled in this study from September 2023 to June 2025, as depicted in Figure 1. In the group of 62 subjects at Pusan National University Hospital, six individuals were excluded for the following reasons: (1) gait dysfunction (n = 4) and (2) inability to complete the evaluation (n = 2). In total, 56 participants (27 diagnosed with ASO and 29 healthy adults) were included in the study. Healthy control participants were recruited from the general community with no history or symptoms of peripheral arterial disease. Their vascular health was verified through detailed medical history review, absence of intermittent claudication, and confirmation of an ABI ≥ 1.00 on both sides. Individuals with borderline ABI values (0.91–0.99) or vascular comorbidities (e.g., diabetes with neuropathy, previous revascularization) were not included to ensure a truly healthy reference group.

The study protocol was approved by the Institutional Review Board of the Pusan National University Hospital (IRB No. 2203-034-112). All participants received a comprehensive explanation of the study protocol, and written informed consent was obtained prior to their involvement, in adherence to the ethical principles outlined in the Declaration of Helsinki. All methodologies complied strictly with the approved study protocol and pertinent guidelines and regulations.

### 2.2. Clinical Measurements

Diagnosis of ASO was confirmed by the presence of lower-limb ischemic symptoms, such as intermittent claudication, numbness, coldness, or non-healing wounds of feet, together with an ankle-brachial index (ABI) ≤ 0.90, in accordance with current international guidelines [1,2]. Most patients were classified as Rutherford category 1–3 (mild-to-moderate claudication), consistent with Fontaine stage II. Patients with critical limb ischemia (Rutherford category ≥ 4 or Fontaine stage III–IV) were excluded from the study. In addition, imaging modalities such as duplex ultrasonography or computed tomography angiography (CTA) were used as confirmatory tools when clinically indicated, but the ABI was adopted as the primary diagnostic criterion for consistency across participants.

### 2.3. Physical Function Measurements

Grip strength was quantitatively assessed in kilograms using a hand dynamometer (TKK 5401; Takei Scientific Instruments, Tokyo, Japan) that was appropriately positioned in the dominant hand of the participant. Throughout the evaluation process, the participants maintained an erect posture with their feet positioned shoulder-width apart, and their elbows completely extended while gazing forward. The SMWT was performed outdoors on a uniform, stable surface conforming to a 30 m straight pathway. The participants were directed to cover the greatest distance possible within a time frame of six min without running or external assistance. The SPPB was measured in the side-by-side position (a), semi-tandem position (b), and tandem position (c) for 10 s each. For positions a and b, the scoring system allocated 0-1 points based on duration held (0 points: <10 s; 1 point: ≥10 s). Position c utilized a 0-1-2 scoring system (0 points: <3 s; 1 point: 3–9.99 s; 2 points: ≥10 s). The cumulative score for the assessment ranges from 0 to 4. To evaluate gait speed, a 6 m course was established, incorporating segments for acceleration, time trials, and deceleration to mitigate bias. The participants were instructed to walk at their customary pace to simulate street walking. The test commenced with the word “go” following a 3 s countdown and concluded upon reaching the 4 m mark. Scoring was delineated as follows: 0 points for non-completion; 1 point for times exceeding 6.52 s; 2 points for times between 4.66 and 6.52 s; 3 points for times between 3.62 and 4.65 s; and 4 points for times below 3.62 s. To evaluate the ability to transition from a seated to standing position, a straight-backed, armless chair with a seat height of 46 cm (slightly higher than commonly used 43 cm standard, but within the acceptable tolerance range reported in previous reliability studies [10,11]) was positioned adjacent to the wall. Participants were instructed to cross their arms and perform the stand-sit action five times as rapidly as possible. The duration from the initial seated posture to the final standing position after the fifth repetition was recorded. The assessment framework is determined by the time taken to complete the task (0 points: >60 s; 1 point: ≥16.70 s; 2 points: 13.70–16.70 s; 3 points: 11.20–13.69 s; 4 points: <11.20 s).

### 2.4. Other Measurements

Participants provided demographic information through a questionnaire. Body mass index (BMI) and skeletal muscle mass (SMM) were assessed using an InBody S10 (InBody S10, InBody Co., Ltd., Seoul, Republic of Korea). The presence of depressive symptoms was assessed utilizing the abbreviated version of the Geriatric Depression Scale in the Korean language (SGDS-K). The Korean version of the Mini-Mental State Examination (K-MMSE) was adopted as a method for assessing cognitive decline. During the implementation of the SGDS-K and K-MMSE, the investigator audibly articulated each item, prompting the participants to provide responses.

### 2.5. Statistical Analysis

Statistical analysis was conducted using SPSS software (version 27, SPSS Inc., Chicago, IL, USA). The study population characteristics are reported as mean ± standard deviation (SD) for continuous variables and frequencies and proportions for categorical variables. Chi-square statistics were used for hypothesis testing as necessary. Pearson’s correlation coefficient was used to assess the correlation between the two variables. Multivariate analysis, ANCOVA was performed using all variables from Table 1 as covariates: age, sex (male), hypertension, diabetes, hyperlipidemia, chronic kidney disease (CKD), smoking, alcohol consumption, height, weight, body mass index (BMI), skeletal muscle mass (SMM), fat mass, body fat percentage, systolic blood pressure (SBP), diastolic blood pressure (DBP), ankle-brachial index (ABI), SGDS, and K-MMSE. The machine learning algorithm employed for the prediction of ASO included logistic regression (LR). In this investigation, we implemented SHAP visualization methodologies to effectively illustrate the significance of various features in relation to model predictions. The analytical procedures were performed using Python (version 3.13.5) on a Windows operating system. The three values used to define ASO for the two groups were compared using receiver operating characteristic (ROC) analysis of the area under the ROC curve (AUC), sensitivity, specificity, and cutoff point. The analyses were conducted using MedCalc for Windows ver. 9.1.0.1 (MedCalc^®^ Corp, Mariakerke Ostend, Belgium). Statistical significance was set at a two-tailed *p*-value < 0.05.

## 3. Results

Table 1 compares the demographic and clinical characteristics of the healthy control group and ASO groups. The control group included 25 males (86.2%) and 4 females (13.8%), with a mean age of 70.76 ± 8.25 years. In the ASO group, there were 23 males (85.2%) and 3 females (14.8%), with a mean age of 69.78 ± 9.12 years.

Table 2 presents a comparison of physical function measures between the groups. Significant differences were observed in the SPPB, gait speed, FTSS, and SMWT, whereas grip strength did not differ significantly between the groups.

The SPPB score was significantly lower in the ASO group (10.56 ± 1.55) compared with controls (11.86 ± 0.44; t = 4.35, *p* < 0.001). Gait speed was also significantly slower in the ASO group (1.06 ± 0.24 m·s^−1^) than in controls (1.34 ± 0.18 m·s^−1^; t = 5.03, *p* < 0.001). The FTSS revealed that patients with ASO required more time to complete the task (12.44 ± 3.62 s) than controls (7.36 ± 0.18 s; t = 6.98, *p* < 0.001). The SMWT distance was significantly shorter in the ASO group (405.39 ± 103.03 m) compared with controls (551.03 ± 70.76 m; t = 6.20, *p* < 0.001). However, grip strength did not differ significantly between the groups (28.44 ± 6.52 kg vs. 30.18 ± 6.11 kg; t = 1.03, *p* = 0.308). Multivariate analysis was significantly difference for Gait speed (*p* < 0.05), FTSS (*p* < 0.01), SMWT (*p* < 0.001).

Random forest analysis (R^2^ = 0.76, RMSE = 0.25) identified FTSS, left ABI, right ABI, SMWT, and gait speed as the five most important predictors of ASO (Figure 2 and Figure 3). The optimal cut-off points for these measures were 13.54 s, 0.78, 1.00, 320.13 m, and 0.79 m·s^−1^.

ROC curve analysis demonstrated excellent diagnostic accuracy for the FTSS (AUC = 0.96), followed by the SMWT (AUC = 0.88) and gait speed (AUC = 0.84; all *p* < 0.001). The sensitivities were 96.30%, 88.89%, and 81.48%, and the specificities were 86.21%, 79.31%, and 82.76% for the FTSS, SMWT, and gait speed, respectively (Figure 4).

## 4. Discussion

The main finding of this study is that, in addition to the ABI, the FTSS, SMWT, and gait speed are important predictors of ASO. Random forest analysis identified meaningful cutoff values for these functional measures, and all three tests demonstrated statistically significant associations with the occurrence of ASO. Furthermore, ROC and AUC analyses demonstrated excellent diagnostic accuracy with high sensitivity and specificity, thereby supporting the validity of the proposed thresholds. These findings highlight the potential of performance-based functional assessments as complementary tools to the ABI for the early detection of functional decline in ASO patients [14].

Peripheral arterial occlusive disease (PAOD) is prevalent in the general population, with reported rates ranging from 0.6% to 10%, and shows a marked increase with advancing age [15]. Among these cases, patients diagnosed with ASO experience a decline in functional capacity due to atherosclerotic processes affecting the peripheral arteries [16], which restrict blood flow to the lower extremities and induce intermittent claudication as the hallmark symptom [17]. In severe cases, cardiovascular complications can lead to mortality due to cardiovascular complications [18]. Intermittent claudication, the most common manifestation of ASO, is characterized by reversible muscular ischemia resulting in cramp-like discomfort and aching sensations during ambulation. These symptoms substantially impair physical activity and walking performance, ultimately leading to a significant deterioration in overall functional capacity.

Previous studies have consistently reported that muscle power, defined as the ability to rapidly generate force against resistance, is closely related to functional performance and the capacity for daily activities. For example, demonstrated in 95 Fontaine II ASO patients that lower-limb muscle power derived from the STS test was significantly correlated with MWD, SMWT, gait speed, stair climbing, and quality-of-life measures, and remained associated with functional improvement following supervised exercise training [9]. This finding was further supported by a recent European consensus statement recommending the use of the STS in the functional assessment of PAD [19]. Collectively, these findings underscore the importance of lower-limb muscle power in vascular populations and provide a basis for extending functional tests beyond correlational analyses to clinically applicable thresholds [14].

Beyond conventional functional tests, recent studies have increasingly applied machine-learning approaches in the context of PAD and ASO. For example, Al-Ramini et al. used gait biomechanical data, including joint kinematics, kinetics, and ground reaction forces, to train neural networks and random forest classifiers, achieving up to 89% accuracy in distinguishing patients with PAD from healthy controls [20], thereby highlighting the feasibility of wearable sensor–based ML applications 4. Similarly, Quan et al. developed a machine learning model with hyperparameter optimization to predict PAD severity, where XGBoost achieved an AUC of approximately 0.92 and identified ABI, inflammatory markers, and diabetes as key features, outperforming conventional cut-off-based approaches [21]. Furthermore, a recent systematic review by Aant et al. synthesized 37 studies and concluded that ML methods consistently outperformed traditional statistics in PAD diagnosis, prognosis, and monitoring but also emphasized the lack of clinically interpretable cutoff values as a critical limitation [19,22]. In this context, the present study contributes by providing ML-derived functional cutoff values for the FTSS, SMWT, and gait speed, with their clinical relevance further reinforced by their excellent sensitivity and specificity in ROC/AUC analyses. This combined approach may partially address the gap between standardized and interpretable thresholds and extend the clinical applicability of functional assessments in ASO.

Unlike previous studies that primarily investigated the correlations between functional measures and muscle power within patient cohorts, the present study recruited a healthy control group for direct comparison, and proposed clinically relevant thresholds through both ML-based analysis and ROC validation. This methodological approach underlines the novelty of our findings, showing that while the FTSS demonstrated the strongest predictive performance, the SMWT and gait speed also yielded significant results, suggesting that these three measures may function as complementary indicators that extend beyond correlational associations to clinically useful thresholds in the diagnostic process of ASO.

In this study, all three performance-based measures—FTSS, SMWT, and gait speed—showed statistically significant differences between the ASO and control groups, indicating impaired lower-limb function among patients with ASO. Among the baseline characteristics, height and alcohol consumption, which showed statistically significant differences between groups, may have potentially influenced the observed differences in functional performance [23]. Several studies have shown that anthropometric characteristics, particularly body height and leg length, can influence not only gait speed but also performance in other functional tests such as the FTSS and SMWT. Shorter individuals tend to require greater joint torque during sit-to-stand movements and have shorter stride length during walking, which may contribute to slower performance [24,25,26]. Nevertheless, in our study, the mean height difference between groups was modest (≈6.4 cm, ~4%), which is below the range generally associated with clinically meaningful differences in these functional outcomes. Alcohol consumption also differed significantly between groups, previous research indicates that moderate drinking has minimal influence on muscle function and gait performance [27,28]. In the present study, the control group showed higher alcohol use but better functional outcomes, suggesting that alcohol intake was unlikely to have biased the observed results. The SGDS scores showed a statistically significant difference between groups, with higher values in the ASO group. Although the mean SGDS scores in the ASO group were below the clinical cutoff for depression (≥6 points on the SGDS-K) [29], they were significantly higher than those of the control group. This suggests that patients with ASO may experience subclinical or mild depressive symptoms that could still influence motivation and perceived physical limitation. Interestingly, although the SPPB includes both gait speed and chair-rise components, its total score did not outperform the individual subtests in discriminating ASO from controls. This may be because the composite scoring system compresses the variability of each subtest by summing across heterogeneous domains. In addition, the use of ordinal rather than continuous values could further reduce sensitivity to subtle functional differences. Consequently, subtle functional differences detected in the continuous FTSS or gait speed measures may be diluted in the aggregated SPPB score. Future studies may consider re-weighting or refining SPPB scoring algorithms to improve its discriminative ability in vascular populations such as ASO.

The FTSS has also been employed as a screening or prognostic tool for populations with other diseases. For instance, a performance time of ≥12 s in older adults has been associated with increased fall risk [13,30], ≥15 s with recurrent falls [13], ≥16 s in COPD patients with poor outcomes and higher mortality risk [31,32], and ≈10.9 s as a diagnostic threshold for sarcopenia [33]. These observations illustrate the broad applicability of the FTSS across different clinical conditions. Extending these findings to the vascular population, the present study identified FTSS cut-off values that could aid in the screening and functional stratification of patients with ASO.

This study had several limitations. First, the small sample size may have inflated the statistical power, limiting the generalizability of the findings. Second, selection bias was possible, as only patients who consented to participate were included, many of whom were highly motivated to complete the tests despite discomfort. Third, the participants were mainly patients with mild-to-moderate disease severity; cutoff values may differ in younger or more severe populations. Fourth, owing to its cross-sectional design, causal relationships between physical function decline and ASO onset cannot be fully established. In addition, the predominance of male participants in this study reflects the higher prevalence of ASO among men, which may limit the extrapolation to female patients. The chair height (46 cm) used for FTSS was slightly higher than the conventional 43 cm standard, which may have resulted in marginally shorter task times; however, this deviation (<5%) is unlikely to have affected the overall outcomes. Given these limitations, this study should be interpreted as exploratory, providing preliminary evidence to guide future research. Future multicenter prospective studies are needed to validate the cutoff values presented here and to examine whether FTSS, SMWT, and gait speed are associated with long-term outcomes such as limb salvage, mortality, and hospital readmission.

## 5. Conclusions

The FTSS, SMWT, and gait speed are simple, feasible, and clinically meaningful performance-based tests that can complement the ABI for the evaluation of PAOD. Among them, the FTSS showed the highest predictive value and diagnostic accuracy. However, incorporating the SMWT and gait speed along with the FTSS may provide a more comprehensive and reliable assessment. Such a combined approach can enable timely diagnosis and intervention as well as objective monitoring of functional decline during recurrence. Furthermore, these measures, when used together, can support patient self-assessment and facilitate clearer communication among healthcare providers, ultimately enhancing the overall evaluation of functional recovery before and after treatment in patients with PAOD.

## Figures and Tables

**Figure 1 healthcare-13-02903-f001:**
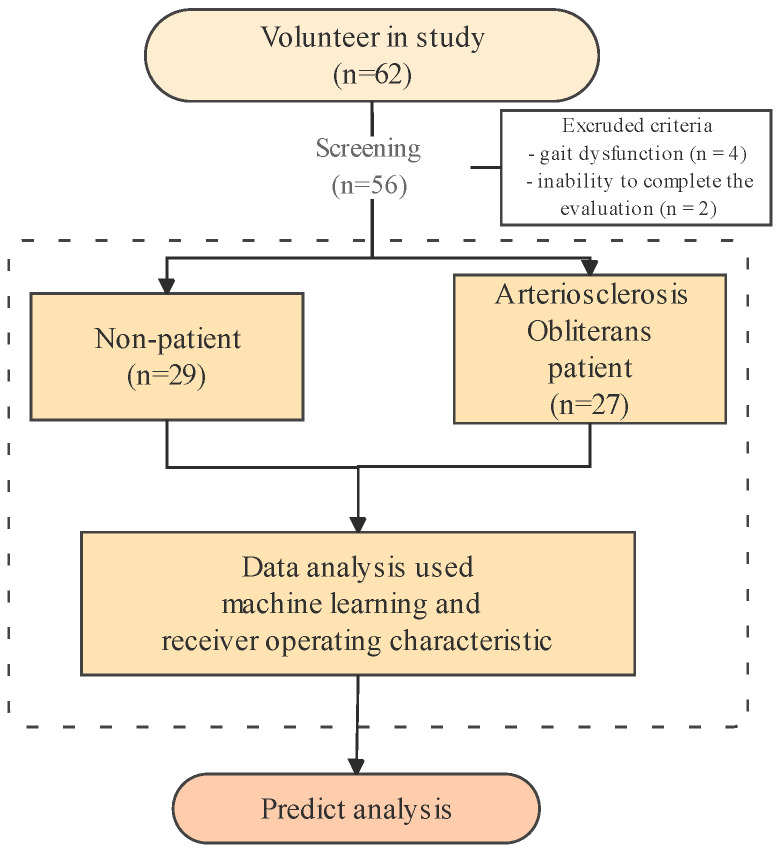
Analysis flow chart in arteriosclerosis obliterans predict analysis.

**Figure 2 healthcare-13-02903-f002:**
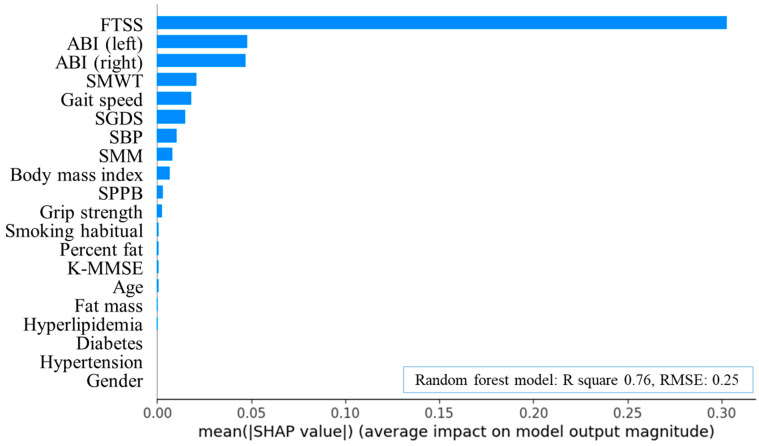
Feature importance plot of random forest algorithms in arteriosclerosis obliterans. FTSS, five times sit-to-stand test; ABI, ankle-brachial index; SMWT, six-minute walk test; SGDS, Short Geriatric Depression Scale; SBP, systolic blood pressure; SMM, skeletal muscle mass; SPPB, short physical performance battery; K-MMSE, Korean version of the Mini-Mental State Examination; R square, coefficient of determination; RMSE, root mean squared error; SHAP, Shapley additive explanations.

**Figure 3 healthcare-13-02903-f003:**
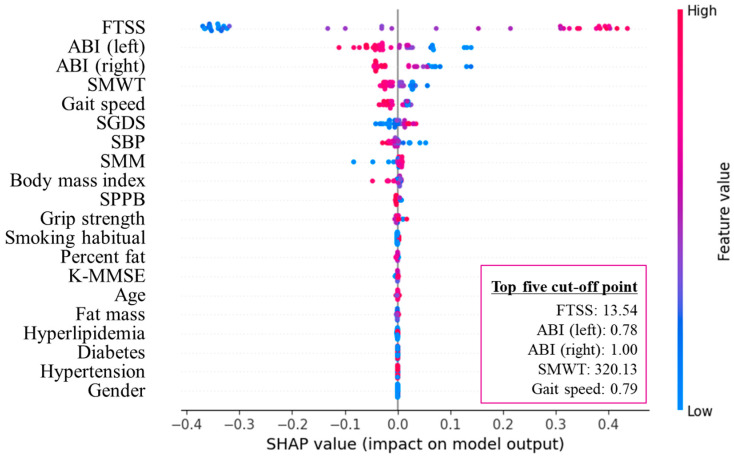
SHAP summary plot of the top five cut-off points in arteriosclerosis obliterans. FTSS, five times sit-to-stand test; ABI, ankle-brachial index; SMWT, six-minute walk test; SGDS, Short Geriatric Depression Scale; SBP, systolic blood pressure; SMM, skeletal muscle mass; SPPB, short physical performance battery; K-MMSE, Korean version of the Mini-Mental State Examination; SHAP, Shapley additive explanations.

**Figure 4 healthcare-13-02903-f004:**
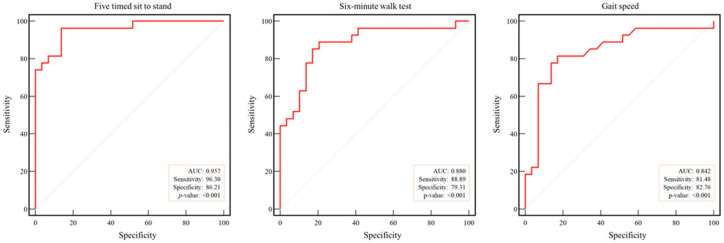
Receiver operating characteristic curve analysis of physical function in arteriosclerosis obliterans. Dotted line: reference; red line: AUC: area under the curve.

**Table 1 healthcare-13-02903-t001:** Characteristics by differences between non-patients and arteriosclerosis obliterans (ASO).

Variables	Non-Patients (n = 29)	ASO (n = 27)	t-Statistic	*p*-Value
Age (years)	70.76 ± 8.25	69.78 ± 9.12	0.42	0.844
Male (%)	25 (86.2%)	23 (85.2%)	-	1.000
Hypertension (%)	12 (41.4%)	18 (66.7%)	-	0.104
Diabetes (%)	7 (24.1%)	11 (40.7%)	-	0.297
Hyperlipidemia (%)	9 (31.0%)	10 (37.0%)	-	0.848
Chronic kidney disease (%)	0 (0%)	4 (14.8%)	-	0.048
Smoking habitual (%)	4 (13.8%)	7 (25.9%)	-	0.421
Alcohol consumption (%)	19 (65.5%)	7 (25.9%)	-	0.007
Height (cm)	161.57 ± 6.46	167.94 ± 7.59	−3.39	0.001
Weight (kg)	63.03 ± 9.49	65.00 ± 7.65	−0.85	0.397
BMI (kg·m^−2^)	24.13 ± 3.51	23.06 ± 2.30	1.34	0.187
SMM (kg)	24.41 ± 4.38	25.89 ± 3.75	−1.35	0.283
Fat mass (kg)	24.59 ± 38.70	17.95 ± 6.36	0.88	0.682
Percent fat (%)	27.61 ± 7.91	27.16 ± 8.84	0.20	0.841
SBP (mmHg)	134.97 ± 17.10	127.93 ± 12.91	1.73	0.090
DBP (mmHg)	74.72 ± 12.37	66.04 ± 13.47	2.52	0.015
ABI (right)	1.12 ± 0.10	0.89 ± 0.23	4.94	<0.001
ABI (left)	1.14 ± 0.12	0.90 ± 0.22	5.23	<0.001
SGDS (point)	1.76 ± 1.64	5.70 ± 3.52	−5.44	<0.001
K-MMSE (point)	27.14 ± 2.26	27.59 ± 1.97	−0.80	0.427

SMM, skeletal muscle mass; BMI, body mass index; SBP, systolic blood pressure; DBP, diastolic blood pressure; ABI, Ankle Brachial Index, SGDS, Short form of Geriatric Depression Scale; K-MMSE, the Korean Mini-Mental State Examination. *p*-value < 0.05.

**Table 2 healthcare-13-02903-t002:** Comparison between non-patients and arteriosclerosis obliterans (ASO) in physical function.

Variables	Non-Patients (n = 29)	ASO (n = 27)	t-Statistic	*p*-Value
SPPB (point)	11.86 ± 0.44	10.56 ± 1.55	4.35	<0.001
Gait speed (m·s^−1^)	1.34 ± 0.18	1.06 ± 0.24	5.03	<0.001 *
FTSS (s)	7.36 ± 1.47	12.44 ± 3.62	−6.98	<0.001 **
Grip strength (kg)	30.18 ± 6.11	28.44 ± 6.52	1.03	0.308
SMWT (m)	551.03 ± 70.76	405.39 ± 103.03	6.20	<0.001 ***

SPPB, short physical performance battery; FTSS, five timed sit to stand; SMWT, six-minute walk test. Adjusted by Age, Male, Hypertension, Diabetes, Hyperlipidemia, CKD, Smoking, Alcohol, Height, Weight, BMI, SMM, Fat, Percent, SBP, DBP, ABI, ABI, SGDS, K-MMSE using ANCOVA. *p*-value < 0.05. * *p* value calculated using ANCOVA, adjusted for covariates (* < 0.05; ** < 0.01; *** < 0.001).

## Data Availability

The datasets analyzed in this study are available from the corresponding author upon request. The data are not publicly available due to privacy and ethical restrictions.

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
