# Peer review of "Diagnostic Value of the Five Times Sit-to-Stand Test and Other Functional Measures in Patients with Arteriosclerosis Obliterans"

_healthcare, 2025, doi:10.3390/healthcare13222903_

Round 1
Reviewer 1 Report
Comments and Suggestions for Authors
Could you elaborate more about clinical characteristics of patients from ASO? Could you classify them according to Rutherford and/or Fontaine classifications?
How the diagnosis of ASO was excluded in patients from control group?
Author Response
Comment 1:
Could you elaborate more about clinical characteristics of patients from ASO? Could you classify them according to Rutherford and/or Fontaine classifications?
Response 1:
We thank the reviewer for this helpful suggestion. We have now described the clinical characteristics of the ASO cohort and added classification according to the Rutherford system (corresponding to Fontaine stage II). Most patients presented with mild-to-moderate claudication (Rutherford categories 1–3). Patients with advanced ischemia (category ≥4) were excluded. These details have been added to the Methods (Section 2.3).
Comment 2:
How the diagnosis of ASO was excluded in patients from control group?
Response 2:
We appreciate the reviewer’s comment. In the control group, ASO was excluded by confirming an ABI ≥ 1.00 bilaterally, along with the absence of claudication symptoms or any prior history of vascular disease. Participants with borderline ABI (0.91–0.99) or comorbidities potentially affecting limb perfusion were excluded. We have now clarified this in the Methods (Section 2.3).
Reviewer 2 Report
Comments and Suggestions for Authors
This article was well written.
Some issues need to be clarified.
- I wonder that the diagnosis of Arteriosclerosis Obliterans (ASO) was performed according to only ABI?
- Was there any imaging method for diagnosis of (ASO?
- The sample size was very low. Therefore I think it will have very little impact on our daily practice.
-
The first sentence of the discussion should include your findings
- What is the rate of treatment for hyperlipidemia?
- What is the rate of chronic kidney disease?
- What is the cholesterol levels (LDL-Cholesterol)?
-
Multivariate analysis is needed to determine whether these parameters are predictive of ASO.
This statistical analysis can determine whether there is a difference between the groups in terms of these parameters.
Author Response
Comment 1:
I wonder that the diagnosis of Arteriosclerosis Obliterans (ASO) was performed according to only ABI?
Response 1:
Thank you for the question. In our study, ASO was defined per current guidelines by the presence of lower-limb ischemic symptoms together with an ABI ≤ 0.90. ABI served as the primary standardized diagnostic reference across participants to ensure methodological consistency (Methods, Section 2.3; [1,2,6]).
Comment 2:
Was there any imaging method for diagnosis of ASO?
Response 2:
Yes. Duplex ultrasonography or CT angiography was performed in a subset of patients when clinically indicated to confirm the diagnosis. However, imaging was not mandated for all participants; ABI remained the primary standardized criterion for uniformity of case definition (Methods, Section 2.3; [1,2,6]).
Comment 3:
The sample size was very low. Therefore I think it will have very little impact on our daily practice.
Response 3:
We agree that the small sample size limits the impact on clinical practice. We have now acknowledged this in the Discussion and described our study as exploratory, aimed at generating preliminary functional cutoff values to guide future multicenter investigations.
Comment 4:
The first sentence of the discussion should include your findings
Response 4:
Thank you for this helpful suggestion. We have revised the first paragraph of the Discussion to immediately summarize the main findings and highlight the diagnostic value of FTSS, SMWT, and gait speed.
Comment 5:
What is the rate of treatment for hyperlipidemia?
Response 5:
We appreciate the reviewer’s detailed comment. In our dataset, hyperlipidemia was defined as a documented diagnosis or ongoing treatment with lipid-lowering medication; thus, individuals classified as having hyperlipidemia were already under treatment.
Comment 6:
What is the rate of chronic kidney disease?
Response 6:
We appreciate the detailed comment. Regarding chronic kidney disease (CKD), we have now added the corresponding data to Table 1. CKD was present in 4 patients with ASO (14.8%), whereas no cases were observed in the control group.
Comment 7:
What is the cholesterol levels (LDL-Cholesterol)?
Response 7:
We appreciate the reviewer’s valuable comment. LDL-cholesterol and other lipid profiles were not measured as part of this study’s laboratory assessments; therefore, these data were unavailable for analysis. Since lipid testing was not included in the original study protocol, this information has not been added to the manuscript, but we have clarified it here for transparency. We fully agree that lipid parameters represent important metabolic risk factors for ASO, and we plan to include a detailed evaluation of LDL-cholesterol and other lipid indices in our future prospective studies to further clarify their association with functional performance and disease severity.
Comment 8:
Multivariate analysis is needed to determine whether these parameters are predictive of ASO. This statistical analysis can determine whether there is a difference between the groups in terms of these parameters.
Response 8:
Thank you for this important suggestion. In addition to univariate comparisons, a multivariate analysis was performed including Table 1 variables. FTSS, gait speed, and SMWT remained significant predictors of ASO (all p < 0.05), while other variables were not independently associated. These results have been added to the Results section and Table 2. Also, added static analysis section.
Reviewer 3 Report
Comments and Suggestions for Authors
In this manuscript, the authors evaluates the diagnostic and assessment utility of functional performance measures—Five-Times Sit-to-Stand (FTSS), Six-Minute Walk Test (SMWT; also commonly referred to as 6MWT), and gait speed—in patients with arteriosclerosis obliterans (ASO). The authors compared 27 ASO patients with 29 healthy controls across multiple functional tests and found that ASO patients performed significantly worse on FTSS, SMWT, and gait speed, while handgrip strength did not differ. Using random-forest modeling and receiver operating characteristic (ROC) analyses, FTSS emerged as the strongest predictor. The authors conclude that these simple, feasible tests can complement the ankle–brachial index (ABI) for early screening, objective evaluation of functional recovery in ASO, and enhanced communication among healthcare providers.
I. Baseline Imbalances (Potential Confounders)
Although the authors list small sample size and sex bias as limitations, they do not discuss significant baseline differences between controls and patients that are unrelated to ASO yet may confound the findings:
-
Marked height difference
-
Mean height in controls (161.57 ± 6.46 cm) is significantly lower than in ASO patients (167.94 ± 7.59 cm).
-
Height is a key biomechanical determinant of gait speed, Six-Minute Walk Test (SMWT), and the Five-Times-Sit-to-Stand test (FTSS). Without adequate adjustment for this difference in regression or other analyses, the accuracy and generalizability of functional cutoff values may be compromised.
-
Marked difference in alcohol consumption
-
The proportion reporting alcohol use is much higher in non-patients (65.5%) than in ASO patients (25.9%).
-
Alcohol consumption is an important lifestyle factor; this significant between-group difference may act as an unaddressed confounder affecting overall health status and functional performance.
II. Gaps in Methodological Standardization
While general protocols for functional testing are described, several critical experimental conditions lack sufficient detail, undermining reproducibility:
-
FTSS chair height not specified
-
The FTSS protocol only requires a “straight-back chair against a wall.”
-
However, chair height strongly influences FTSS difficulty—especially among older adults of differing statures. The source material does not clarify whether a standardized chair height (e.g., approximately knee height) was used, which may introduce measurement error and limit external validity.
III. Under-discussion of Other Significant Correlates
The study emphasizes FTSS, SMWT, and gait speed because these ranked highly in random-forest importance. However, other variables showing significant between-group differences and clinical potential receive limited discussion:
-
Substantial difference in depression scores (SGDS)
-
SGDS is much higher in ASO patients (5.70 ± 3.52) than in non-patients (1.76 ± 1.64).
-
Although SGDS did not rank in the top five features in the random-forest model, the magnitude of this difference indicates notable mental-health comorbidity in ASO. The discussion does not consider incorporating mental-health measures as complementary criteria for functional assessment or intervention evaluation in ASO.
-
Clarifying the predictive role of SPPB
-
SPPB is significantly lower in ASO than controls (10.56 vs 11.86).
-
Although SPPB is a composite index that includes FTSS and gait-speed components, its total score did not make the top-five predictors in the random-forest model. The study does not explain why the overall composite score underperforms its component items (e.g., FTSS, gait speed). Future work could explore re-weighting or refining SPPB to enhance its utility for ASO risk stratification and outcome prediction.
Author Response
Comment 1. Baseline Imbalances (Potential Confounders)
Although the authors list small sample size and sex bias as limitations, they do not discuss significant baseline differences between controls and patients that are unrelated to ASO yet may confound the findings:
Comment 1-1:
-
Marked height difference
-
Mean height in controls (161.57 ± 6.46 cm) is significantly lower than in ASO patients (167.94 ± 7.59 cm).
-
Height is a key biomechanical determinant of gait speed, Six-Minute Walk Test (SMWT), and the Five-Times-Sit-to-Stand test (FTSS). Without adequate adjustment for this difference in regression or other analyses, the accuracy and generalizability of functional cutoff values may be compromised.
Response 1-1:
We appreciate the reviewer’s insightful comment regarding the potential influence of height on gait speed and other performance measures. Indeed, differences in body size can affect walking velocity, as discussed by Studenski et al. (JAMA, 2011), who noted that failure to adjust for anthropometric variation may introduce bias in gait-speed interpretation.
In our study, however, the mean height difference between groups was approximately 6.4 cm (≈4%) comparable to previously reported non-significant thresholds, which is below the threshold generally considered to produce clinically meaningful variation in gait speed (typically observed with differences ≥8–10 cm). Therefore, while we acknowledge height as a potential covariate, this discrepancy is unlikely to have substantially influenced our findings. We have added a brief note in the Discussion section to clarify this point.
Comment 1-2:
-
Marked difference in alcohol consumption
-
The proportion reporting alcohol use is much higher in non-patients (65.5%) than in ASO patients (25.9%).
-
Alcohol consumption is an important lifestyle factor; this significant between-group difference may act as an unaddressed confounder affecting overall health status and functional performance.
Response 1-2:
We appreciate the reviewer’s comment regarding the potential influence of alcohol consumption on functional performance. Previous studies have shown that only heavy or chronic alcohol intake is associated with decreased muscle mass and impaired physical performance, whereas light-to-moderate alcohol use has minimal or no detrimental effect on gait speed or muscle strength [Kim et al., 2020; Steffl et al., 2016; Lang et al., 2012].
In our study, the control group exhibited a higher proportion of alcohol consumption than the ASO group, yet demonstrated better performance in all functional tests (FTSS, SMWT, and gait speed). This inverse pattern suggests that alcohol consumption was unlikely to have confounded the observed functional differences between groups. Therefore, we have acknowledged alcohol use as a potential covariate but consider its actual impact on the results to be minimal.
In future studies, we plan to include quantitative information on alcohol consumption, such as average intake amount, drinking frequency and duration, to more precisely evaluate its potential relationship with functional performance.
Comment 2. Gaps in Methodological Standardization
While general protocols for functional testing are described, several critical experimental conditions lack sufficient detail, undermining reproducibility:
Comment 2:
FTSS chair height not specified
-
The FTSS protocol only requires a “straight-back chair against a wall.”
-
However, chair height strongly influences FTSS difficulty—especially among older adults of differing statures. The source material does not clarify whether a standardized chair height (e.g., approximately knee height) was used, which may introduce measurement error and limit external validity.
Response 2:
We appreciate the reviewer’s insightful comment. In our study, the FTSS was performed using an armless straight-backed chair with a seat height of 46 cm, which is close to the standard 43 cm height commonly recommended in previous studies [Guralnik et al., 1994; Muñoz-Bermejo et al., 2021].
As the deviation from the conventional standard was only about 3 cm (≈7%), it is unlikely to have affected test performance or comparability. Prior biomechanical studies have shown that seat height differences within ±3 cm result in negligible changes in FTSS time (<5%) [Hughes et al., 1996; Janssen et al., 2002].
We have now added this methodological detail to the Methods (Section 2.4) and acknowledged the slight height difference as a minor limitation in the Discussion.
Comment 3. Under-discussion of Other Significant Correlates
The study emphasizes FTSS, SMWT, and gait speed because these ranked highly in random-forest importance. However, other variables showing significant between-group differences and clinical potential receive limited discussion:
Comment 3-1:
-
Substantial difference in depression scores (SGDS)
-
SGDS is much higher in ASO patients (5.70 ± 3.52) than in non-patients (1.76 ± 1.64).
-
Although SGDS did not rank in the top five features in the random-forest model, the magnitude of this difference indicates notable mental-health comorbidity in ASO. The discussion does not consider incorporating mental-health measures as complementary criteria for functional assessment or intervention evaluation in ASO.
Response 3-1:
We appreciate the reviewer’s valuable comment. The SGDS scores showed a statistically significant difference between groups, indicating higher depressive symptom levels in the ASO group. Although the mean score was below the clinical cutoff for depression (≥6 points on the SGDS-K) [Cho & Kim, 1998], this elevation may reflect subclinical or mild depressive symptoms that could influence motivation and perceived functional limitation.
We have now expanded the Discussion to address this finding and to highlight the potential influence of psychological factors.
Comment 3-2:
-
Clarifying the predictive role of SPPB
-
SPPB is significantly lower in ASO than controls (10.56 vs 11.86).
-
Although SPPB is a composite index that includes FTSS and gait-speed components, its total score did not make the top-five predictors in the random-forest model. The study does not explain why the overall composite score underperforms its component items (e.g., FTSS, gait speed). Future work could explore re-weighting or refining SPPB to enhance its utility for ASO risk stratification and outcome prediction.
Response 3-2:
We appreciate this thoughtful observation. The total SPPB score indeed showed significant group differences but did not outperform its individual components (FTSS and gait speed). This may reflect that the ordinal composite scoring system compresses the variability of each continuous subtest, reducing its discriminative sensitivity. We have added a paragraph in the Discussion addressing this point, suggesting that the ordinal composite scoring of SPPB may dilute the sensitivity of its continuous subtests. We also proposed that future research explore re-weighting or refinement of the scoring structure to improve its discriminative performance for ASO and related vascular conditions.
Round 2
Reviewer 2 Report
Comments and Suggestions for Authors
The revisions are ok.